# Untargeted LC-MS/MS Metabolomics Study on the MCF-7 Cell Line in the Presence of Valproic Acid

**DOI:** 10.3390/ijms23052645

**Published:** 2022-02-28

**Authors:** Alan Rubén Estrada-Pérez, Martha Cecilia Rosales-Hernández, Juan Benjamín García-Vázquez, Norbert Bakalara, Benedicte Fromager, José Correa-Basurto

**Affiliations:** 1Laboratorio de Diseño y Desarrollo de Nuevos Fármacos e Innovación Biotecnológica (Laboratory for the Design and Development of New Drugs and Biotechnological Innovation), SEPI, Escuela Superior de Medicina, Instituto Politécnico Nacional, Plan de San Luis y Díaz Mirón, Ciudad de Mexico 11340, Mexico; alan.estrada1375@gmail.com (A.R.E.-P.); benjagv_5202@hotmail.com (J.B.G.-V.); 2Laboratorio de Biofísica y Biocatálisis, SEPI, Escuela Superior de Medicina, Instituto Politécnico Nacional, Plan de San Luis y Díaz Mirón, Ciudad de Mexico 11340, Mexico; 3CNRS, ENSTBB-Bordeaux INP, Univeristé de Bordeaux, 146 rue LéoSaignat, 33000 Bordeaux, France; norbert.bakalara@bordeaux-inp.fr; 4Ecole Nationale Supérieure de Chimie de Montpellier, 34296 Montpellier, France; benedicte.fromager@enscm.fr

**Keywords:** breast cancer, MCF-7, HDAC8, VPA, metabolomics analysis

## Abstract

To target breast cancer (BC), epigenetic modulation could be a promising therapy strategy due to its role in the genesis, growth, and metastases of BC. Valproic acid (VPA) is a well-known histone deacetylase inhibitor (HDACi), which due to its epigenetic focus needs to be studied in depth to understand the effects it might elicit in BC cells. The aim of this work is to contribute to exploring the complete pharmacological mechanism of VPA in killing cancer cells using MCF-7. LC-MS/MS metabolomics studies were applied to MCF-7 treated with VPA. The results show that VPA promote cell death by altering metabolic pathways principally pentose phosphate pathway (PPP) and 2′deoxy-α-D-ribose-1-phosphate degradation related with metabolites that decrease cell proliferation and cell growth, interfere with energy sources and enhance reactive oxygen species (ROS) levels. We even suggest that mechanisms such as ferropoptosis could be involved due to deregulation of L-cysteine. These results suggest that VPA has different pharmacological mechanisms in killing cancer cells including apoptotic and nonapoptotic mechanisms, and due to the broad impact that HDACis have in cells, metabolomic approaches are a great source of information to generate new insights for this type of molecule.

## 1. Introduction

Breast cancer (BC) is the most malignant tumor found in women and is located in the terminal ductal lobular units of epithelial tissue. BC is the most common cancer that affects women worldwide, with more than one million cases per year [1]. The most common breast cancer molecular subtype is luminal A (according to a retrospective study conducted in Saudi Arabia) and in consequence is the most important subtype [2]. BC is a multifactorial disease associated with genetic and environmental factors [3]. To be more efficient against BC, several therapeutic strategies are applied: radiotherapy, surgery, chemotherapy, and adjuvant therapy. Although chemotherapy is currently one of the most common treatments for BC, there are several side effects [4] because chemotherapy is not focused against localized cancer targets or cell surfaces and/or intracellular signaling pathways [5].

It is known that epigenetic changes are involved in carcinogenesis, cancer growth, and metastasis [6]. Included in these epigenetic changes are histone acetylation/deacetylation at Lys residues. Histones are important targets that lead to either decondensation or condensation of the chromatin, affecting gene transcription [7]. Histone acetylation and deacetylation at Lys residues are mainly controlled by histone acetyltransferases (HAT) and histone deacetylases (HDAC), respectively [8]. Histone acetylation exposes the genes to transcription factors that inhibit cell survival, decrease cell migration and favor cell apoptosis, etc. [9].

Many cellular functions are regulated by the acetylation/deacetylation of histones [10] and nonhistone proteins [11]. Thus, the inhibition of HDAC increases the protein acetylation/deacetylation ratio, which has anticancer effects. In fact, HDAC inhibitors (HDACi) induce cancer cell cycle arrest, cell death, a decrease in angiogenesis and other biological responses. Therefore, HDAC inhibitors have been studied and developed as promising anticancer agents [12].

The biological effects of HDACi depend on the type of cancer as well as on the targeted HDAC isoform [13]. There are 18 HDAC isoforms that are divided into four classes [14]. From these 18 HDAC isoforms, HDAC1, 6 and 8 isoform are overexpressed in BC; therefore, these HDAC isoforms may be specific biological targets for BC [15]. However, HDACi can inhibit other HDAC isoforms associated with cell death as well as target both histone and nonhistone proteins associated with different cell processes (e.g., energy homeostasis) that perturb the cancer cells [16].

Valproic acid (VPA) (Figure 1) is a HDACi used as an epileptic seizure suppressor and a mood stabilizer. It is known that VPA has an antiproliferative effect on many cancer cells, including BC cells, by inhibiting the activity of HDACs [17].

In this work, we focused on extending the current knowledge about effect of VPA on MCF-7 cells using untargeted LC-MS/MS metabolomics studies to identify the intracellular pathways that could suggest the possible full pharmacological mechanism of action related to cell death.

## 2. Results and Discussion 

### Metabolomics Analysis of MCF-7 Treated with VPA

Metabolomics is a powerful tool for studying upregulated or downregulated metabolites that are related to intracellular pathways, which helps in understanding the mechanisms of action that explain a drug’s pharmacological effects [18]. In this work, we have explored VPA using metabolomic studies to decipher its effect in MCF-7 metabolome and cell death. 

The LC-MS/MS analyses show 2583 different features between untreated MCF-7 cells versus treated MCF-7 cells with VPA. However, 74 metabolites were significantly deregulated (Figure 1). Features were mainly highly hydrophobic in nature since they present high retention in the reverse-phase separation system. Most features were downregulated (53 features) and 12 were identified through METLIN database (Table 1). This study shows that VPA mainly downregulates metabolites that influence some intracellular pathways (Figure 2) mainly involved in energetics and redox control. Putative metabolites for each significantly dysregulated pathway are also shown (Table 2). Here, we describe some of these deregulated metabolites found in MCF-7 due to their relationship with cell death. 

VPA upregulates glycocholate, and on the contrary, downregulates 7α and 12α-dihydroxycholest-4-en-3-one which are involved in the lower and upper part, respectively, of the bile acid biosynthesis pathway that leads to the production of bile acids [19]; glycocholate is also related to drug resistance in cancer cells [20]. Additionally, it is known that bile acid conjugates favor thymidine incorporation in MCF-7 cells [21]. Bile acids are also associated with tumorigenesis via complex mechanisms including oxidative stress, which damages DNA, induces apoptosis and modulates epigenetic factors that affect gene expression with reduced and/or increased expression of nuclear receptors [22]. Moreover, it has been shown that the upregulation of the bile acid biosynthesis pathway induces apoptosis [23]. It is known that HDAC inhibitors overexpress CYP7A1, an enzyme that regulates the bile acid biosynthesis pathway [24].

VPA treatment downregulates two of the products of transketolase (TKT) enzyme, which is involved in the nonoxidative branch of the pentose phosphate pathway (PPP): D-sedoheptulose 7-phosphate and D-glyceraldehyde 3-phosphate. PPP is a relevant pathway in cancer development because mutated cells tend to show upregulation, particularly in genes encoding ribose-5-phosphate isomerase (RPI) and ribulose-5-phosphate epimerase (RPE), to fulfill their energetic and amino acid and nucleic acid demand due to their fast division requirements [25,26,27]. Downregulation of PPP elements might be responsible for MCF-7 cell death by depleting their energetic capabilities and limiting their cell capacity to synthesize more nucleotides for DNA repairing. Complementarily, not only TKT but also transaldolase (TALDO1) concentrations are slightly increased in HL60 cells treated with VPA, although their acetylation ratio increases too [28].

The most dysregulated pathway was 2′-deoxy-α-D-ribose 1-phosphate degradation, with both metabolites identified to be downregulated. It is worth mentioning that D-glyceraldehyde 3-phosphate is a metabolite shared between this and PPP, so its effect might be more related to this last pathway. Alterations in PPP, pyrimidine, purine, and primary bile acids metabolism were also found by Zhou et al. (2019) using a quadrupole-orbitrap instrument in MCF-7 cells, so this is a consistent effect exerted by VPA in this cell line. Besides alterations in energetic metabolism, these authors also found alterations in amino acid metabolism in this cell line and MDA-MB-231, both from breast cancer [29].

Besides the alterations in PPP, the downregulation of L-cysteine attracted our attention because in recent years the importance of this amino acid has increased with our increasing knowledge of cancer metabolism rewiring [30,31]. As Serpa (2020) indicated, cysteine has a pivotal role in several metabolic adjustments that a cancer cell must make, mainly in three areas: redox control, ATP production and carbon source. About this last insight, cysteine is also related to PPP, since it works as a gluconeogenesis precursor, which ultimately develops into glucose-6-phosphate, a substrate for PPP in both oxidative and nonoxidative branches. Analysis of cysteine through the predictive metabolic results allows us to reinforce the importance of this amino acid in the effect of VPA due to its participation in several metabolic pathways such as coenzyme biosynthesis, cysteine biosynthesis and degradation, hydrogen sulfide biosynthesis and homocysteine degradation, serotonin, and taurine biosynthesis, and others. Cell redox capacity is another worth mentioning subject, Contis-Montes de Oca et al. (2018) found that HO-VPA, a novel compound partly based on VPA which shares many of its effects, increased ROS levels (through HMB1 acetylation) and decreased cell viability in HeLa cells [32,33]. Reduction in cysteine availability, a known limiting factor in GSH synthesis, may play a key role in cell death through glutathione shortage [30]. Coincidently, one of the most dysregulated metabolites found by Zhou et al. (2019) was L-cysteine, just falling behind (R)-pantothenic acid in significance level, although they found it to be upregulated. According to novel studies, a decrease in intracellular cysteine may lead to ferroptosis, a term thatr refers to a nonapoptotic mechanism of cell death, which involves accumulation of lipid peroxides and is also closely related to a reduction in glutathione [34,35].

Downregulation of β-L-fucose 1-phosphate might be associated with an increase in GDP-β-L-fucose, and although we did not register such a change in this last molecule, it has been discussed that increasing L-fucose levels inhibit cell growth in vitro. Such an increase can translate into more fucosylated glycans with tumor-suppressive properties, which may lead to an eventual depletion in the levels of β-L-fucose 1-phosphate [36].

17 β-estradiol is an estrogenic hormone associated with cell proliferation and migration in different cancer types, BC among them (especially hormone-dependent cell lines such as MCF-7), and whose response involves estrogen receptors’ stimulation by this hormone [37,38,39]. Blocking or disruption of the stimulation process may lead to inhibition of this cancer hallmark [40], so 17 β-estradiol decrease as a consequence of VPA presence might be involved in the effect observed in inhibition of cell proliferation. Another metabolite involved in BC inhibition was stearate, which has been proved to inhibit diverse processes in cancer proliferation that ultimately lead to apoptosis [38,41]. Particularly, evaluations on stearate’s role in apoptosis induction, as well as cell migration and invasion, showed that it selectively affects BC cells (MDA-MB-435 and Hst578t) over breast cells (MCF-10A and Hs578Bst) as its concentration increases [42]. Here, we found upregulation of stearate because of VPA treatment, which supports previously found evidence that this compound elicits apoptosis, evaluated in cancer stem cells, and this outcome may be related to increased concentration of stearate as a consequence of VPA presence [43].

Special attention is focused on nicotinamide, a precursor of NAD^+^ (nicotinamide adenine dinucleotide) which is directly involved in many cancer process such as redox reactions, DNA repair through poly(ADP-ribose) polymerase (PARP) and gene expression through sirtuins [44,45]. Downregulation of nicotinamide may alter the outcome of the aforementioned processes and it needs further investigation to understand the role of this dysregulation in VPA´s antiproliferative effect in MCF-7.

In summary, VPA is a HDACi that, although it has been thoroughly studied, many studies had been focused on its toxic metabolites. Therefore, this work is important because it shows that VPA modified some metabolic pathways, as was recently reported, that VPA inhibits mitochondrial bioenergetics [46], which is in accordance with the information reported now in this research. Therefore, metabolic studies still offer new insights as new or different technologies are used to generate data that help to characterize VPA’s effect in breast cancer.

## 3. Material and Methods

### 3.1. Cell Culture and Treatment

The MCF-7 cells (considered luminal A subtype) [47] were cultured in Dulbecco’s Modified Eagle Medium (DMEM, Gibco, Ciudad de México, México), supplemented with 5% decomplemented fetal bovine serum (FBS, Biowest, Kansas City, MO, USA). The cell cultures were maintained at 37 °C in a humidified atmosphere with 5% CO_2_. For the experiments, the cells were seeded into a Petri dish, and incubated for 6 days to obtain approximately 7,000,000 cells per dish. The MCF-7 cells were treated with VPA using its reported IC_50_ value of 0.74 mM for VPA [48] for 48 h. Each treatment was made in triplicate and a blank was included in the analysis. 

### 3.2. Steps for Metabolites Extraction

*Step 1: The Petri dish was placed on wet ice and the DMEM was collected to remove the exometabolites to measure only the intracellular metabolites. The cells were washed twice with ice-cold PBS (DPBS) and frozen on dry ice. MeOH:H_2_O (2:0.8) (kept on dry ice) was added to the dish that was then transferred to wet ice before scrapping the cells. The cells were collected into an Eppendorf tube and snap-frozen in liquid nitrogen to be stored at −80 °C until processing in step 2.

*Step 2: The samples were sonicated, one part chloroform was added for a total solution ratio of 2:0.8:1 of MeOH:H_2_O:CHCl_3_, and the samples were then vortexed. Next, one volume of water was added, and the samples were vortexed again. One volume of CHCl_3_ was then added, followed by vortexing and centrifugation (30 min, 5000 rpm, 4 °C). Aqueous and organic phases were separated, dried down and stored at −80 °C. The organic phase was analyzed by UHPLC-MS/MS.

#### 3.2.1. UHPLC-MS/MS Data Acquisition

For the UHPLC-Q-TOF-MS studies, an Agilent 1290 Infinity II system coupled with a 6545A Q-TOF with a dual AJS ESI source (Agilent Technologies, Santa Clara, CA, USA) was used. The flow rate of the mobile phase was 0.5 mL/min with an injection volume of 10 μL for each sample. Separation of organic phase was performed in an Agilent Eclipse XDB-C8, 4.6 × 150 mm^2^, 5 µm column using two solvents: formic acid 0.1% (A) and formic acid 0.1% in methanol LC-MS grade (B). The gradient elution program went from 10 to 100% of B over 20 min, followed by 100% B maintained for 5 min. The column temperature throughout the separation process was kept at 40 °C. During all LC-MS/MS analyses, samples were kept in an autosampler at 4 °C.

The ESI source was operated in positive ion mode with the following conditions: the nebulizer gas pressure was set at 35 psig, Nozzle voltage was set at 500 V, the capillary voltage was set at 3500 V, drying gas flow rate and temperature were set at 10 L/min and 300 °C, sheath gas flow rate and temperature were set to 10 L/min and 280 °C, respectively and fragmentor voltage was set at 120 V. For MS/MS measurements, a collision energy ramp was used to promote fragmentation with a slope of 3 and an offset of 15, selecting 5 precursors per cycle using Auto MS/MS acquisition mode. The data was acquired with MassHunter Workstation LC/MS Data Acquisition B.08.00 Software in centroid and profile mode using High Resolution mode (4 GHz). The mass range was set at 100–1000 m/z in MS and 50–1000 m/z in MS/MS mode. A scan rate of 3 spectra/sec was used in both cases. All data were acquired using Agilent MassHunter Workstation Software LC/MS Data Acquisition for 6200 series TOF/6500 series Q-TOF, version B.08.00, build 8.00.8058.3 SP1 (Agilent Technologies, Santa Clara, CA, USA).

#### 3.2.2. LC-MS/MS Analysis

The raw data obtained by UHPLC-Q-TOF-MS/MS were converted into mzData through Agilent MassHunter Workstation Software Qualitative Analysis, version B.07.00, build 7.7.7024.29 SP2 (Agilent Technologies, Santa Clara, CA, USA) to be analyzed with XCMS Online version 3.7.1 (www.xcmsonline.scripps.edu (accessed on 10 June 2020)) [49]. Information was analyzed by pairwise analysis: MCF-7 untreated cells versus MCF-7 treated with VPA. The parameters selected for the jobs were those available for a UPLC/UHD Q-TOF, which included: m/z tolerated deviation of 15 ppm, peak width from 5 to 20 s, signal/noise threshold of 6, mzdiff of 0.01, mzwid of 0.015, bw of 5, minfrac and minsamp of 0.5 and 1, respectively, a fold change of 1.5, *p*-value threshold of 0.05 for significant features and 0.01 for highly significant features. An unpaired parametric *t*-test (Welch *t*-test) was used to evaluate the results. Putative metabolite annotation and identification were performed through the METLIN database available through XCMS, allowing a 5 ppm error and 0.015 m/z absolute error for the annotation and a 10 ppm tolerance for the database search using human as the selected biosource for the identification [50]. Only those features that were identified through this method were further considered as putative metabolites.

## 4. Conclusions

This work shows that VPA can address several biological targets through HDAC inhibition, because this compound deregulates several intracellular metabolites involved in pathways significant to cancer cell survival. Molecules and pathways previously reported are confirmed in this study, but new compounds are also reported, which helps to enlighten the mechanism through which VPA exerts it effect in BC cells.

## Data Availability

The data presented in this study are openly available in FigShare associated to DOI 10.6084/m9.figshare.19090268.

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
