# Peer review of "Untargeted LC-MS/MS Metabolomics Study on the MCF-7 Cell Line in the Presence of Valproic Acid"

_ijms, 2022, doi:10.3390/ijms23052645_

Round 1
Reviewer 1 Report
The authors present a novel approach to analyze the pharmacological effects of valproic acid on the breast cancer cell line, MCF7. The study is mainly observational and lacks an in-depth mechanistic insight that further validates the use of LC-MS/MS in conducting such analysis. Several approaches to improve and strengthen this study can be employed including:
1- The use of additional breast cancer cell lines that represent various breast cancer subsets
2- Investigation of whether the changes can be detected in a dose-response manner rather than just testing a previously reported IC50
3- Performing further mechanistic studies to determine whether the observed effects on the metabolites reported are due to epigenetic regulation, which would further validate the methodology utilized in the study
Author Response
Referee 1
The authors present a novel approach to analyze the pharmacological effects of valproic acid on the breast cancer cell line MCF-7. The study is mainly observational and lacks in-depth mechanistic insight that further validate the use of LC-MS/MS in conducting such analysis. Several approaches to improve and strengthen this study can be employed including:
- The use of additional breast cancer cell lines that represent various breast cancer subsets.
Response: Thank you for your comments and we agree that breast cancer, as many other types of cancer, is a quite challenging subject which involves different molecular subtypes that might be represented by different cell lines. We have added a couple of lines (36 - 38) to highlight the importance of molecular subtype luminal A, which is the most common and indicated in line 183 that MCF-7 is consider into luminal A subtype (the corresponding references were added and updated). So now is possible to see that we focused on this main type of cancer subtype. We also consider increasing the number of cell lines to test in further studies. Thank you for calling this into our attention.
- Investigation of whether the changes can be detected in a dose-response manner ratter than just testing a previously reported IC50.
Response: Thank you for the observation. Indeed, a dose-response study could provide new and different insights about VPA effect, but we decided to focus on a concentration in which we could be able to observe an effect elicit by VPA presence, authors like Del-Coco et al. (2020) have also used this approach to ensure a cellular response: https://www.ncbi.nlm.nih.gov/pmc/articles/PMC7435671/. Also, we selected this concentration to gain insights and to enrich our previous work on VPA (https://pubmed.ncbi.nlm.nih.gov/27483122/).
- Performing further mechanistic studies to determine whether the observed effects on the metabolites reported are due to epigenetic regulation, which would further validate the methodology utilized in the study.
Response: At that moment we were interested in finding compounds that showed a differential response to VPA, so now that we have found some of this, we may be able to design other study that focuses on these compounds. We really appreciate your suggestion.
Reviewer 2 Report
The proposed study is very comprehensive and with a clear goal.
In this study, the authors investigated the complete pharmacological mechanism of VPA in killing cancer cells using MCF-7. The manuscript includes interesting information which revealed that VPA promotes cell death by altering metabolic pathways, mainly the pentose phosphate pathway (PPP) and the degradation of 2´deoxy-α-D-ribose-1-phosphate concerning metabolites that reduce cell proliferation and growth, interfere with energy sources and increase ROS levels. Also, the authors confirm that VPA has different pharmacological mechanisms in killing cancer cells, including apoptotic and non-apoptotic mechanisms.
In general, the manuscript is well written and revised, so the reviewer does not find errors in this version of the manuscript.
All research activities were performed in detail.
Statistical processing of the results and their categorization were also systematically performed.
Material and methods: Very well written.
Results: This section is excellently written with all the necessary and concise accompanying explanations.
The results correspond to the objectives of the study.
In the tables, all results are well presented and easy to compare.
The figures are in a satisfactory resolution so that all the mentioned and explained details are visible.
Discussion: The discussion part is included in the results and explained to a completely satisfactory extent.
The references used are carefully selected and also up-to-date.
Author Response
Response: The referee 2 has no comments or suggestions. We are grateful for all his compliments.
Round 2
Reviewer 1 Report
The authors addressed my previous comments.
This manuscript is a resubmission of an earlier submission. The following is a list of the peer review reports and author responses from that submission.
Round 1
Reviewer 1 Report
The manuscript entitled "Untargeted LC-MS metabolomics study on the MCF-7
cell line in the presence of HO-AAVPA and VPA" intends to provide information about metabolic changes occurring in cancer cells after treating them with VPA and HO-AAVPA. In my opinion, the manuscript must be improved and it is not suitable for publications in its current form. The following comments should be addressed:
- The title should have the full name of the drugs that are going to be tested.
- The authors use the terms features and metabolites as if they could be interchanged. Several features can correspond to one single metabolite. This is not acceptable in a metabolomics paper.
- It is not clear what was the criteria to state that 2583 and 12271 features were found. Features should be filtered based on blanks background detected features, and the use of a pooled QC that should be run multiple times during samples instrumental analysis.
- It is not clear how the identity of the metabolites was confirmed. Did the authors perform MS/MS experiments? did they use standards? what confidence do the authors have to claim that those metabolites were actually changing?
- A table listing the significant metabolites is missing. Such table should include p-values, retention times, fold changes, pathway affected, information about identification, etc.
- Figures 1 and 3 don't provide important information to the paper.
- Line 128, page 4, a reference there is missing.
- Page 5, line 135, did you observe changes in glutamate.
- I don't understand why the authors report nicotine, bupropion, and aspirine pathways if these substances were not added to the cell cultures.
- The discussion should be more grounded and less speculative. The authors elaborate complex conclusions without having enough evidence of the metabolic changes occurring within the cells. If you observe a metabolite being up-regulated or down-regulated you can discuss the possible pathways that could be affected and elaborate something about how that could be related to cancer. However, a lot of details implying that if a given metabolite increases, then another metabolite X should be de-regulated, and then because of that reason another pathway should be affected, without actually having evidence that the metabolite X is actually changing. This is too much speculation.
- was the extraction workflow previously optimized? how do you know the effectiveness of this strategy? is there any reference where such workflow was reported?
- Why there is no data in negative mode?
- what type of univariate test was done? parametric or non-parametric?
- How many replicates per treatment were analyzed?
Reviewer 2 Report
This manuscript described a untargeted metabolomics study to investigate MCF-7 cell line in the presence of HO-AAVPA and VPA. Overall, this is a relatively straightforward study with simple experimental design. The conclusion that HO-AAVPA and VPA have different pharmacological mechanisms in killing cancer cells is supported by the data.
My main comments are on the lack of details in describing their data analysis and the presentation should also be improved. Unlike other omics, untargeted metabolomics data analysis is not well-established and authors should be cautious in using different methods and describe them with sufficient details. Just mentioning XCMSonline or Agilent Software are not very meaningful
More specific comments are given below:
1) Methods/parameters for data processing (peak detection, missing values, etc) and normalization (log, unit scaling, etc) - this is not mentioned but critical for data analysis
2) Statistical analysis p values - from t-tests? How did it handle multi-testings?
3) How compound identification and pathway analysis were performed?
4) How to interpret the Fig 2, Fig 4, Fig 5.
5) In the results discussion, the authors are very facile in discussing the up-/down-regulation of different compounds and involvement of different pathways. It is much better to have a table summary to highlight the key pathways and metabolite hits, p values to show the supporting evidence.
6) It may be desirable to present a chart showing the possible mechanisms